# Eukaryotic CD-NTase, STING, and viperin proteins evolved via domain shuffling, horizontal transfer, and ancient inheritance from prokaryotes

**Edward M. Culbertson, Tera C. Levin** *

University of Pittsburgh, Department of Biological Sciences, Pittsburgh, Pennsylvania, United States of America

* teralevin@pitt.edu

## Abstract

Animals use a variety of cell-autonomous innate immune proteins to detect viral infections and prevent replication. Recent studies have discovered that a subset of mammalian antiviral proteins have homology to antiphage defense proteins in bacteria, implying that there are aspects of innate immunity that are shared across the Tree of Life. While the majority of these studies have focused on characterizing the diversity and biochemical functions of the bacterial proteins, the evolutionary relationships between animal and bacterial proteins are less clear. This ambiguity is partly due to the long evolutionary distances separating animal and bacterial proteins, which obscures their relationships. Here, we tackle this problem for 3 innate immune families (CD-NTases [including cGAS], STINGs, and viperins) by deeply sampling protein diversity across eukaryotes. We find that viperins and OAS family CD-NTases are ancient immune proteins, likely inherited since the earliest eukaryotes first arose. In contrast, we find other immune proteins that were acquired via at least 4 independent events of horizontal gene transfer (HGT) from bacteria. Two of these events allowed algae to acquire new bacterial viperins, while 2 more HGT events gave rise to distinct superfamilies of eukaryotic CD-NTases: the cGLR superfamily (containing cGAS) that has since diversified via a series of animal-specific duplications and a previously undefined eSMODS superfamily, which more closely resembles bacterial CD-NTases. Finally, we found that cGAS and STING proteins have substantially different histories, with STING protein domains undergoing convergent domain shuffling in bacteria and eukaryotes. Overall, our findings paint a picture of eukaryotic innate immunity as highly dynamic, where eukaryotes build upon their ancient antiviral repertoires through the reuse of protein domains and by repeatedly sampling a rich reservoir of bacterial antiphage genes.

**Data Availability Statement:** All relevant data are within the paper and its Supporting Information files. Additional code used in the paper is available

at https://github.com/MBL-Physiology-Bioinformatics/2021-Bioinformatics-Tutorial-Materials/tree/master/phylogenetics

**Funding:** This research was supported in part by the University of Pittsburgh Center for Research Computing, RRID:SCR_022735, through the resources provided. Specifically, this work used the HTC cluster, which is supported by NIH award number S10OD028483. EMC was supported by NSF Postdoctoral fellowship 2208971 and TCL was supported by NIH R00AI139344 and R35GM150681. The funders had no role in study design, data collection and analysis, decision to publish, or preparation of the manuscript.

**Competing interests:** The authors have declared that no competing interests exist.

**Abbreviations:** blSTING, bacteria-like STING; CBASS, cyclic oligonucleotide-based antiphage signaling system; CD-NTase, cGAS-DncV like nucleotidyltransferase; cGAS, cyclic GMP-AMP synthase; HGT, horizontal gene transfer; HMM, hidden Markov model; LECA, last eukaryotic common ancestor; PAP, Poly(A) RNA polymerase; STING, Stimulator of Interferon Genes.

## Introduction

As the first line of defense against pathogens, all forms of life rely on cell-autonomous innate immunity to recognize threats and respond with countermeasures. Until recently, many components of innate immunity were thought to be lineage-specific [1]. However, new studies have revealed that an ever-growing number of proteins used in mammalian antiviral immunity are homologous to bacterial immune proteins used to fight off bacteriophage infections. This list includes Argonaute, CARD domains, cGAS and other CD-NTases, Death-like domains, Gasdermin, NACHT domains, STING, SamHD1, TRADD-N domains, TIR domains, and viperin, among others [2–13]. Perhaps one of the most exciting discoveries from these bacterial defense systems is the highly varied biochemical functions carried out by these bacterial proteins. For example, bacterial cGAS-DncV-like nucleotidyltransferases (CD-NTases), which generate cyclic nucleotide messengers (similar to cGAS), are massively diverse with over 6,000 CD-NTase proteins discovered to date. Beyond the cyclic GMP-AMP signals produced by animal cGAS proteins, bacterial CD-NTases are capable of producing a wide array of nucleotide signals including cyclic dinucleotides, cyclic trinucleotides, and linear oligonucleotides [11,14]. Many of these bacterial CD-NTase products are critical for bacterial defense against viral infections [8]. Interestingly, these discoveries with the CD-NTases mirror what has been discovered with bacterial viperins. In mammals, viperin proteins restrict viral replication by generating 3′-deoxy-3′,4′didehdro- (ddh) nucleotides [4,15–17], which block RNA synthesis and thereby inhibit viral replication [15,18]. Mammalian viperin generates ddhCTP molecules while bacterial viperins can generate ddhCTP, ddhUTP, and ddhGTP. In some cases, a single bacterial protein is capable of synthesizing 2 or 3 of these ddh derivatives [4]. These discoveries have been surprising and exciting, as they imply that some cellular defenses have deep commonalities spanning across the entire Tree of Life, with additional new mechanisms of immunity waiting to be discovered within diverse microbial lineages. But despite significant homology, these bacterial and animal immune proteins are often distinct in their molecular functions and operate within dramatically different signaling pathways (reviewed here [5]). How, then, have animals and other eukaryotes acquired these immune proteins?

One common hypothesis in the field is that these immune proteins are ancient and have been inherited since the last common ancestor of bacteria and eukaryotes [5]. In other cases, horizontal gene transfer (HGT) between bacteria and eukaryotes has been invoked to explain the similarities [6,19]. However, because most papers in this field have focused on searching genomic databases for new bacterial immune genes and biochemically characterizing them, the evolution of these proteins in eukaryotes has not been as thoroughly investigated.

We investigated the ancestry of 3 gene families that are shared between animal and bacterial immunity: Stimulator of Interferon Genes (STING), cyclic GMP-AMP synthase (cGAS) and its broader family of CD-NTases, and viperin. STING, CD-NTases, and viperin are all interferon-stimulated genes that function as antiviral immune modules, disrupting the viral life cycle by activating downstream immune genes, sensing viral infection, or disrupting viral processes, respectively [20]. We choose to focus on the cGAS, STING, and viperin for a number of reasons. First, in metazoans cGAS and STING are part of the same signaling pathway, whereas bacterial CD-NTases often act independently of bacterial STINGs [21], raising interesting questions about how eukaryotic immune proteins have gained their signaling partners. Also, given the vast breadth of bacterial CD-NTase diversity, we were curious as to if any eukaryotes had acquired CD-NTases distinct from cGAS. For similar reasons, we investigated viperin, which also has a wide diversity in bacteria but a much more narrow described function in eukaryotes.

We found eukaryotic CD-NTases arose following multiple HGT events between bacteria and eukaryotes. cGAS fall within a unique, mainly metazoan clade. In contrast, OAS-like proteins were independently acquired and are the predominant type of CD-NTase found across most eukaryotes. Separately, we have discovered diverged eukaryotic STING proteins that bridge the evolutionary gap between metazoan and bacterial STINGs, as well as 2 separate instances where bacteria and eukaryotes have acquired similar proteins via convergent domain shuffling. Finally, we find that viperin was likely present in the LECA and possibly earlier, with both broad representation across the eukaryotic tree of life and evidence of 2 additional HGT events where eukaryotes recently acquired new bacterial viperins. Overall, our results demonstrate that immune proteins shared between bacteria and eukaryotes are evolutionarily dynamic, with eukaryotes taking multiple routes to acquire and deploy these ancient immune modules.

## Results

### Discovering immune homologs across the eukaryotic tree of life

The first step to understanding the evolution of CD-NTases, STINGs, and viperins was to acquire sequences for these proteins from across the eukaryotic tree. To search for diverse immune homologs, we employed a hidden Markov model (HMM) strategy, which has high sensitivity, a low number of false positives, and the ability to separately analyze multiple (potentially independently evolving) domains in the same protein [22–24]. We used this HMM strategy to search the EukProt database, which has been developed to reflect the true scope of eukaryotic diversity through the genomes and transcriptomes of nearly 1,000 species, specifically selected to span the eukaryotic tree [25]. EukProt contains sequences from NCBI and Ensembl, plus many diverged eukaryotic species not found in any other database, making it a unique resource for eukaryotic diversity [25]. While it can be challenging to acquire diverse eukaryotic sequences from traditional databases due to an overrepresentation of metazoan data [26], EukProt ameliorates this bias by downsampling traditionally overrepresented taxa.

To broaden our searches from initial animal homologs to eukaryotic sequences more generally, we used iterative HMM searches of the EukProt database, incorporating the hits from each search into the subsequent HMM. After using this approach to create pan-eukaryotic HMMs for each protein family, we then added in bacterial homologs to generate universal HMMs (Figs 1A and S1), continuing our iterative searches until we either failed to find any new protein sequences or began finding proteins outside of the family of interest (S1 Fig). To define the boundaries that separated our proteins of interest from neighboring gene families, we focused on including homologs that shared protein domains that defined that family (see Materials and methods for domain designations) and were closer to in-group sequences than the outgroup sequences on a phylogenetic tree (outgroup sequences are noted in the Materials and methods).

Our searches for CD-NTases, STINGs, and viperins recovered hundreds of eukaryotic proteins from each family, including a particularly large number of metazoan sequences (red bars, Fig 1B). It is not surprising that we found so many metazoan homologs, as each of these proteins was discovered and characterized in metazoans and these animal genomes tend to be of higher quality than other taxa (S2 Fig). We also recovered homologs from other species spread across the eukaryotic tree, demonstrating that our approach could successfully identify deeply diverged homologs (Fig 1B). However, outside of Metazoa, these homologs were sparsely distributed, such that for most species in our dataset (711/993), we did not recover proteins from any of the 3 immune families examined (white space, lack of colored bars, Fig 1B). While some of these absences may be due to technical errors or dataset incompleteness (S2 Fig), we interpret this pattern as a reflection of ongoing, repeated gene losses across eukaryotes, as has been

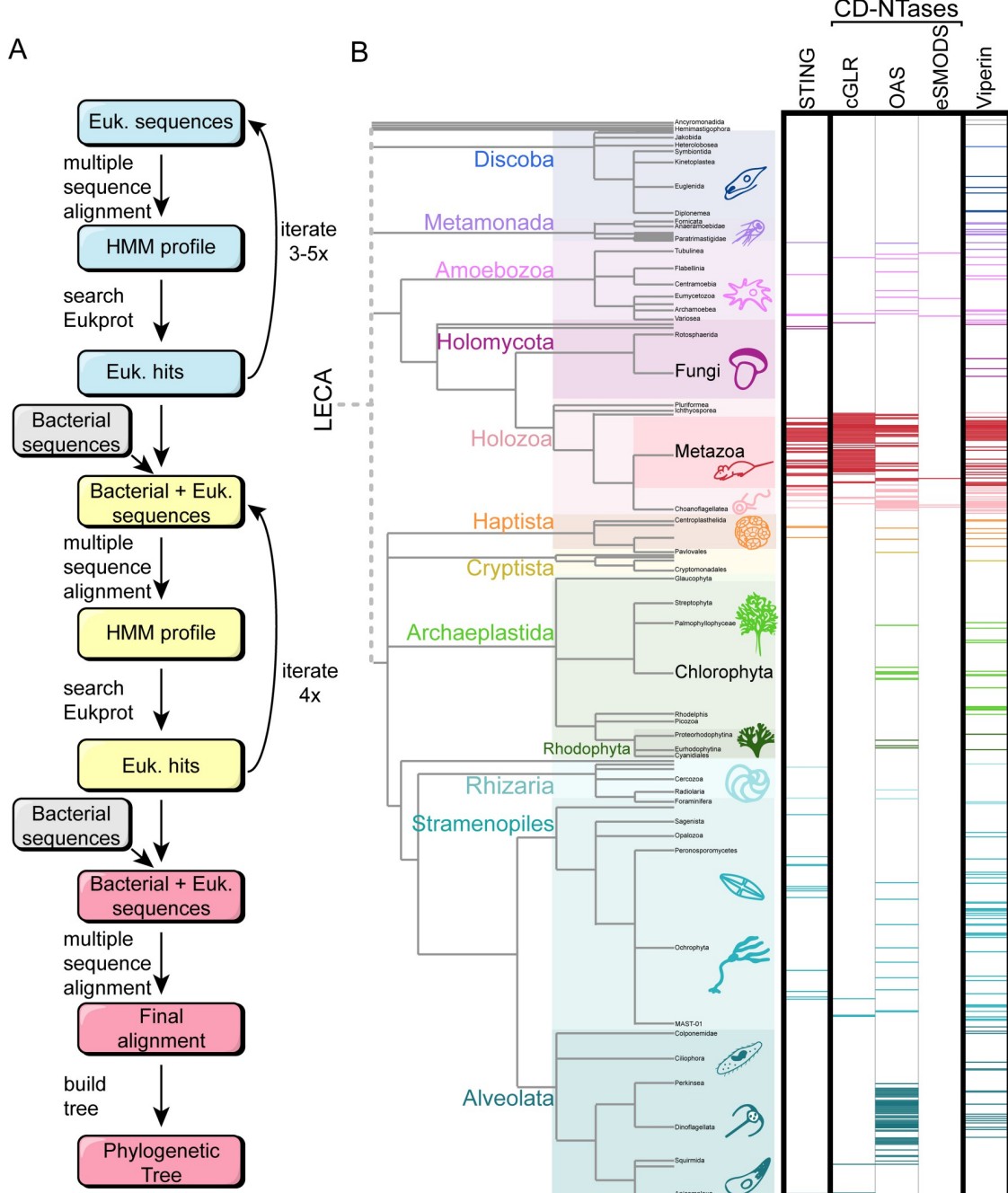

**Fig 1. HMM searches to find homologs across the eukaryotic Tree of Life.** (A) A schematic of the HMM search process. Starting from initial, animal-dominated HMM profiles for each protein family, we used iterative HMM searches of the EukProt database to generate pan-eukaryotic HMMs. These were combined with bacterial sequences to enable discovery of bacteria-like homologs in eukaryotes. Each set of searches was repeated until few or no additional eukaryotic sequences were recovered which was between 3 and 5 times in all cases. (B) Phylogenetic tree of eukaryotes, with major supergroups color coded. The height of the colored rectangles for each group is proportional to its species representation in EukProt. Horizontal, colored bars mark each eukaryotic species in which we found homologs of STINGs, CD-NTases, or viperins. White space indicates species where we searched but did not recover any homologs. The CD-NTase hits are divided into the 3 eukaryotic superfamilies, defined in Fig 2. Individual data are available in S1 File. CD-NTase, cGAS-DncV-like nucleotidyltransferase; HMM, hidden Markov model; STING, Stimulator of Interferon Genes.

found for other innate immune proteins [27–29] and other types of gene families surveyed across eukaryotes [28,30–32]. Indeed, many of the species that lacked any of the immune homologs were represented by high-quality datasets (Ex: Metazoa, Chlorplastida, and Fungi). Thus, although it is always possible that our approach has missed some homologs, we believe the resulting data represents a fair assessment of the diversity across eukaryotes, at least for those species currently included within EukProt.

## Eukaryotes acquired CD-NTases from bacteria through multiple, independent HGT events

We next studied the evolution of the innate immune proteins, beginning with cGAS and its broader family of CD-NTase enzymes. Following infections or cellular damage, cGAS binds cytosolic DNA and generates cyclic GMP-AMP (cGAMP) [33–36], which then activates downstream immune responses via STING [35,37–39]. Another eukaryotic CD-NTase, 2′5′-Oligoadenylate Synthetase 1 (OAS1), synthesizes 2′,5′-oligoadenylates that bind and activate Ribonuclease L (RNase L) [40]. Activated RNase L is a potent endoribonuclease that degrades both host and viral RNA species, reducing viral replication (reviewed here [41,42]). Some bacterial CD-NTases such as *DncV* behave similar to animal cGAS; they are activated by phage infection and produce cGAMP [8,21,43]. These CD-NTases are commonly found within cyclic oligonucleotide-based antiphage signaling systems (CBASS) across many bacterial phyla and archaea [8,21,44].

In addition to the well-studied cGAS, a number of other eukaryotic CD-NTases have been previously described: the OAS1 paralogs (OAS2/3), Male abnormal 21-Like 1/2/3/4 (MAB21L1/2/3/4), Mab-21 domain containing protein 2 (MB21D2), Mitochondrial dynamics protein 49/51(MID49/51), and Inositol 1,4,5 triphosphate receptor-interacting protein 1/2 (ITPRILP/1/2) [44]. Of these, cGAS and OAS1 are the best characterized and both play roles in immune signaling. Recent work has shown that cGAS and related animal proteins, the cGAS-like Receptors (cGLRs), are present in nearly all metazoan taxa and generate diverse cyclic dinucleotide signals [45]. However, the immune functions of Mab21L1 and MB21D2 remain unclear, although Mab21L1 has been shown to be important for development [46–48].

To analyze the evolutionary history of the eukaryotic CD-NTases, we searched EukProt v3 for homologs and then generated phylogenetic trees. We aligned the homologs with MAFFT and MUSCLE and then generated phylogenetic trees with IQtree and RaxML (see Materials and methods). We considered our results to be robust if they were concordant across the majority of 4 trees generated per gene.

To begin our sequence searches for eukaryotic CD-NTases, we used the Pfam domain PF03281, representing the main catalytic domain of cGAS, as a starting point. As representative bacterial CD-NTases, we used 6,132 bacterial sequences, representing a wide swath of CD-NTase diversity [21]. Following our iterative HMM searches, we recovered 313 sequences from 109 eukaryotes, of which 34 were metazoans (S30, S31 and S32 Files and Fig 1B). Within the phylogenetic trees, most eukaryotic sequences clustered into one of 2 distinct superfamilies: the cGLR superfamily (defined by clade and containing a Mab21 PFAM domain: PF03281) or the OAS superfamily (OAS1-C: PF10421) (Fig 2A). Bacterial CD-NTases typically had sequences matching the HMM for the Second Messenger Oligonucleotide or Dinucleotide Synthetase domain (SMODS: PF18144).

The cGLR superfamily is composed almost entirely of metazoan sequences, with only a few homologs from Amoebozoa, choanoflagellates, and other eukaryotes (Fig 2A). Indeed, the majority of animal CD-NTases (cGAS, Mid51, Mab21, Mab21L1/2/3/4, Mb21d2, ITPRI) are paralogs within the cGLR superfamily, which arose from repeated animal-specific duplications

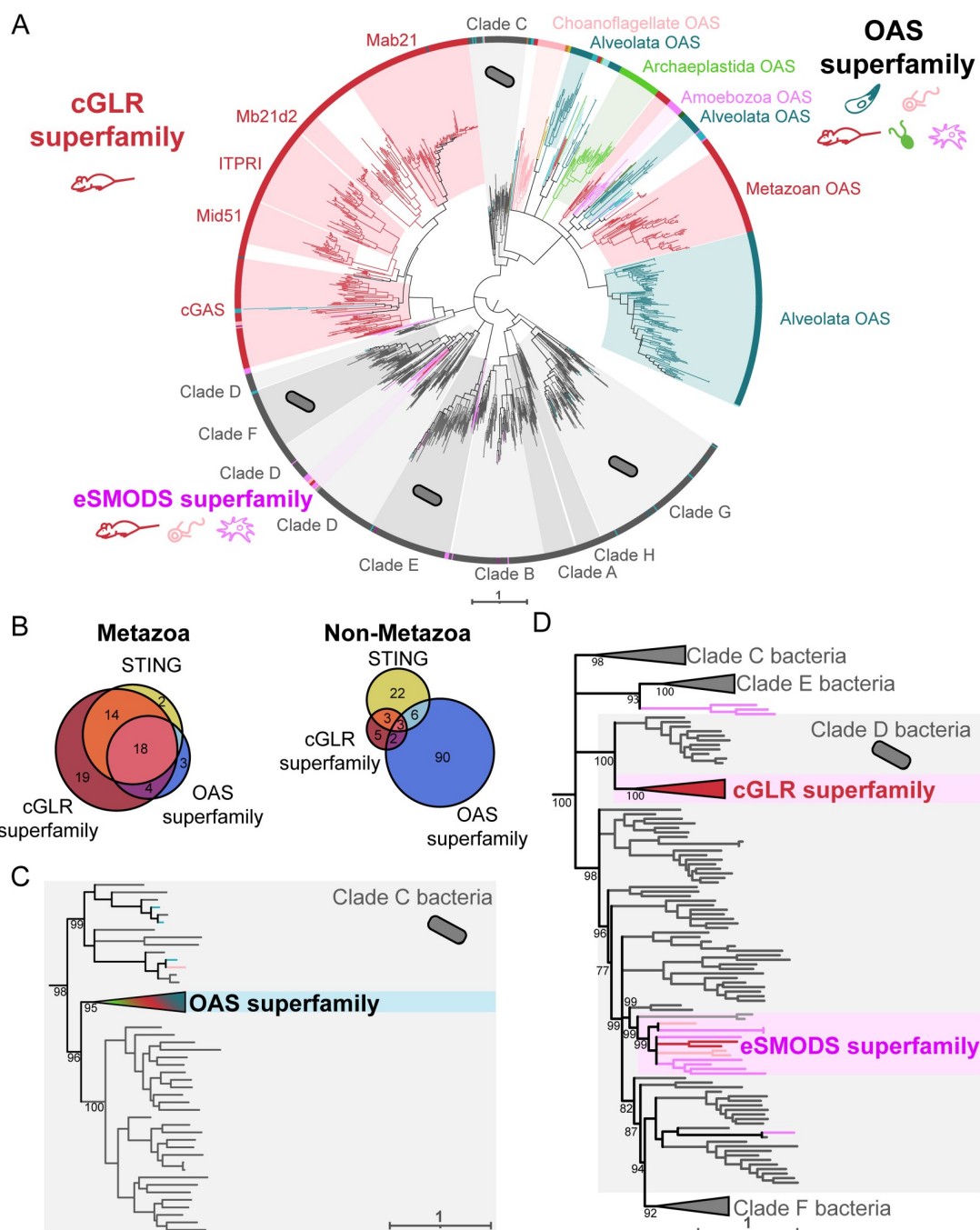

**Fig 2. Independent HGT events gave rise to multiple CD-NTase superfamilies.** (A) Maximum likelihood phylogenetic tree generated by IQtree of CD-NTases spanning eukaryotic and bacterial diversity. The cGLR superfamily (red, top left) is largely an animal-specific innovation, with many paralogs including cGAS. In contrast, most other eukaryotic lineages encode CD-NTases from the OAS superfamily (multicolor, top right). The relatively small eSMODS superfamily (pink, bottom left) likely arose from a recent HGT between clade D bacteria and eukaryotes. Bacterial CD-NTase sequences shown in gray. Eukaryotic sequences are colored according to eukaryotic supergroup as in Fig 1B. Tree is arbitrarily rooted on a branch separating bacterial clades A, B, G, and H from the rest of the bacterial CD-NTases. (B) Venn diagrams showing the number of species where we detected at least 1 STING, cGLR, and/or OAS homolog, either within Metazoa (left) or in non-metazoan eukaryotes (right). (C) Magnification of the CD-NTase phylogenetic tree in (A), showing the region where the OAS superfamily branches within clade C bacterial CD-NTases (gray branches). (D) Magnification showing clade D CD-NTases (gray branches), which have been horizontally transferred into eukaryotes multiple times, giving rise to both the cGLR and the eSMODS superfamilies. Ultrafast bootstraps determined by IQtree shown at key nodes. See S4 Fig for full CD-NTase phylogenetic tree. Underlying Newick file is included in S2 File. Additional information on which species encode CD-NTases of a given homolog (Fig 2B) can be found in S1 File.

CD-NTase, cGAS-DncV-like nucleotidyltransferase; cGAS, cyclic GMP-AMP synthase; HGT, horizontal gene transfer; STING, Stimulator of Interferon Genes.

[49] (S4 Fig). In contrast, unlike the animal-dominated cGLR superfamily, the OAS superfamily spans a broad group of eukaryotic taxa, with OAS-like homologs present in 8/12 eukaryotic supergroups. This distribution makes OAS proteins the most common CD-NTases found across eukaryotes and implies that they arose very early in eukaryotic history, possibly before the LECA.

Given the connections between cGAS and STING in both animals and some bacteria [3,21,50], we asked whether species that encode STING also have cGLR and/or OAS proteins. Because the cGLR superfamily is largely animal specific, we performed this analysis separately in either Metazoa or with all non-metazoan eukaryotes (Fig 2B). In animal species where we found a STING homolog, we also typically found a cGLR superfamily sequence (32/34), and specifically a cGAS homolog (26/34 species) (Fig 2B), consistent with the consensus that these proteins are functionally linked. We also observed 19 metazoan species that had a cGLR-like sequence with no detectable STING homolog. Almost half of these species (10/19) were arthropods, aligning with prior findings of STING sparseness among arthropods [50]. We did find STING homologs in 8/19 arthropod species in EukProt v3, including the previously identified STINGs of *Drosophila melanogaster*, *Apis mellifera*, and *Tribolium castaneum* [50,51]. Outside of animals, we found that species with a STING homolog typically did not have a detectable CD-NTase protein from either superfamily (22/34). While it remains possible that these STING proteins function together with a to-be-discovered CD-NTase that was absent from our dataset, we therefore hypothesize that many eukaryotes outside of metazoans and their close relatives [52] use STING and CD-NTase homologs independently of each other.

What was the evolutionary origin of eukaryotic CD-NTases? Interestingly, the cGLR and OAS superfamilies are only distantly related to one another. Each lies nested within a different, previously defined, bacterial CD-NTase clade (Fig 2C and 2D). The OAS superfamily falls within bacterial Clade C (with the closest related bacterial CD-NTases being those of subclade C02-C03, Fig 2C), while the metazoan cGLR superfamily lies within bacterial Clade D (subclade D12) (Fig 2D). We note that in this tree (Fig 2D), Clade D does not form a single coherent clade, as was also true in the phylogeny that originally defined the bacterial CD-NTase clades [11].

We also observed a number of eukaryotic sequences scattered across different bacterial CD-NTase clades (Fig 2A, colored branches within gray clades). While some of these may reflect additional HGT events, others likely come from technical artifacts such as bacterial contamination of eukaryotic sequences. To minimize such false positive HGT calls, we took a conservative approach in our analyses, considering potential bacteria–eukaryote HGT events to be trustworthy only if: (1) eukaryotic and bacterial sequences branched adjacent to one another with strong support (bootstrap values >70); (2) the eukaryotic sequences formed a distinct subclade, represented by at least 2 species from the same eukaryotic supergroup; (3) the eukaryotic sequences were produced by at least 2 different studies; and (4) the position of the horizontally transferred sequences was robust across all alignment and phylogenetic reconstruction methods used (S3A Fig). For species represented only by transcriptomes, these criteria may still have difficulty distinguishing eukaryote–bacteria HGT from certain specific scenarios such as the long-term presence of dedicated, eukaryote-associated, bacterial symbionts. However, because these criteria allow us to focus on relatively old HGT events, they give us higher confidence these events are likely to be real.

The cGLR superfamily passed all 4 of the HGT thresholds, as did another eukaryotic clade of CD-NTases that were all previously undescribed. We name this clade the eukaryotic SMODS (eSMODS) superfamily, because the top scoring domain from hmmscan for each sequence in this superfamily was the SMODS domain (PF18144), which is typically found only in bacterial CD-NTases (S25 File). This sequence similarity suggests that eSMODS arose following a recent HGT from bacteria and/or that these CD-NTases have diverged from their bacterial predecessors less than the eukaryotic OAS and cGLR families have. Additionally, all of the eSMODS sequences were predicted to have a Nucleotidyltransferase domain (PF01909), and (8/12) had a Polymerase Beta domain (PF18765), which are features shared with many bacterial CD-NTases in Clades D, E, and F (S25 File). The eSMODS superfamily is made up of sequences from Amoebozoa, choanoflagellates, Ancryomonadida, and 1 animal (the sponge *Oscarella pearsei*), which clustered together robustly and with high support (ultrafast bootstrap value of 99) within bacterial Clade D (e.g., subclade D04, CD-NTase 22 from *Myxococcus xanthus*) (S4 Fig). The eSMODS placement on the tree was robust to all alignment and phylogenetic algorithms used (S3A Fig), suggesting that eSMODS represent an additional, independent acquisition of CD-NTases from bacteria.

CD-NTases from bacterial Clade C and Clade D are the only CD-NTases to produce cyclic trinucleotides, producing cyclic tri-Adenylate and cAAG, respectively [11,14,53,54]. Interestingly, OAS produces linear adenylates, which is one step away from the cAAA product made by previously characterized Class C CD-NTases, and similarly cGAMP (made by cGAS) is one adenylate away from the Clade D product cAAG. As of this writing, the Clade D CD-NTases closest to the eSMODS and cGLR superfamilies (D04 and D12, respectively) have not been well characterized. Therefore, we argue that these CD-NTases should be a focus of future studies, as they may hint at the evolutionary stepping stones that allow eukaryotes to acquire bacterial immune proteins.

## Diverged eukaryotic STINGs bridge the gap between bacteria and animals

We next turned to analyze STING proteins. In animals, STING is a critical cyclic dinucleotide sensor, important during viral, bacterial, and parasitic infections (reviewed here [55]). Structurally, most metazoan STINGs consist of an N-terminal transmembrane domain (TM), made of 4 alpha helices fused to a C-terminal STING domain [56]. Canonical animal STINGs show distant homology with STING effectors from the bacterial CBASS, with major differences in protein structure and pathway function between these animal and bacterial defenses. For example, in bacteria, the majority of STING proteins are fusions of a STING domain to a TIR (Toll/interleukin-1 receptor) domain (Fig 3A). Bacterial STING proteins recognize cyclic di-GMP and oligomerize upon activation, which promotes TIR enzymatic activity [3,57,58]. Some bacteria, such as *Flavobacteriaceae*, encode proteins that fuse a STING domain to a transmembrane domain, although it is unclear how these bacterial TM-STINGs function [3]. Other bacteria have STING domain fusions with deoxyribohydrolase, α/β- hydrolase, or trypsin peptidase domains [19]. In addition to eukaryotic TM-STINGs, a few eukaryotes such as the oyster *Crassostrea gigas* have TIR-STING fusion proteins, although the exact role of their TIR domain remains unclear [3,51,59].

Given these major differences in domain architectures, ligands, and downstream immune responses, how have animals and bacteria evolved their STING-based defenses, and what are the relationships between them? Prior to this work, the phylogenetic relationship between animal and bacterial STINGs has been difficult to characterize with high support [19]. Indeed, when we made a tree of previously known animal and bacterial STING domains, we found

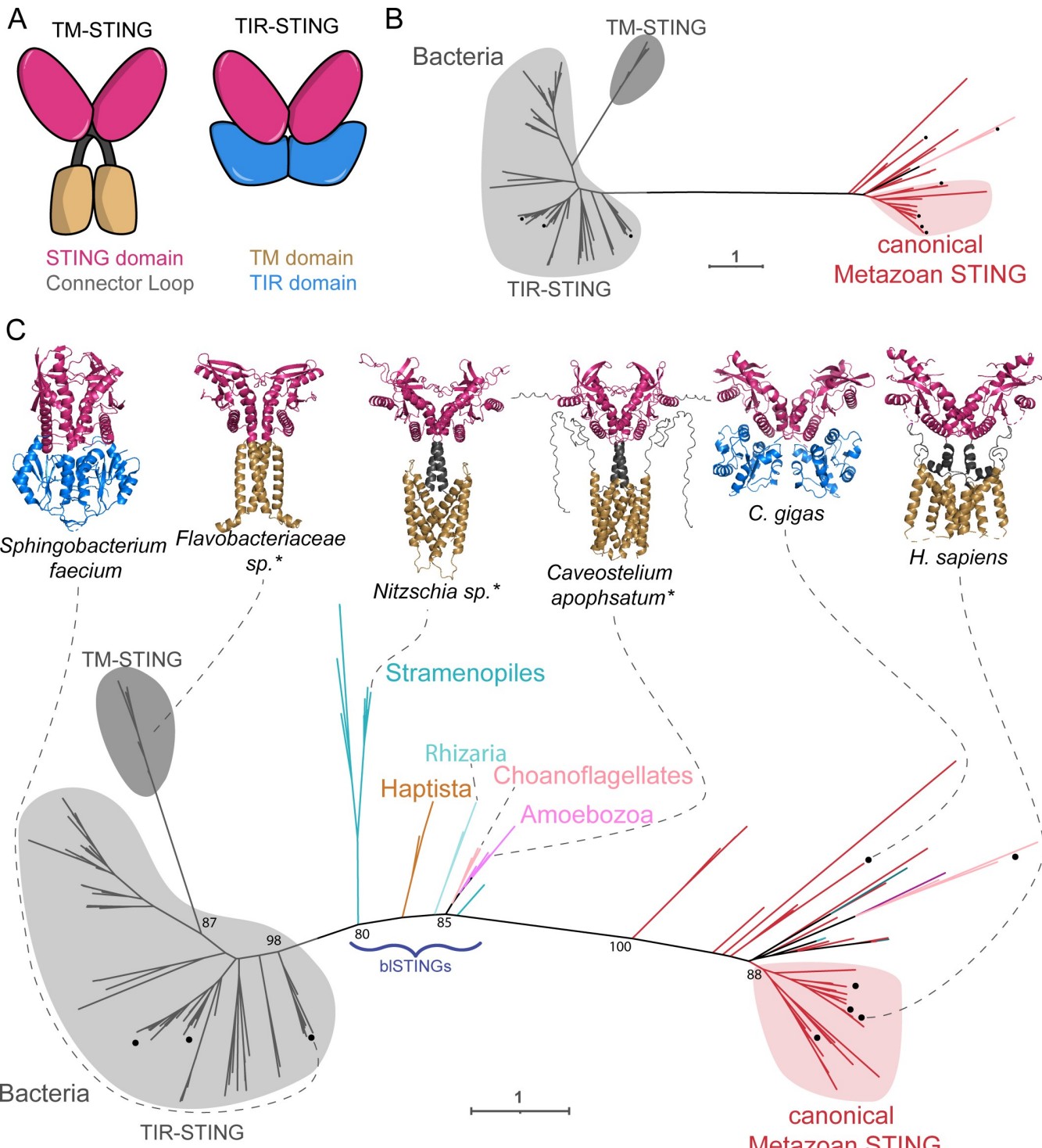

**Fig 3. Diverse eukaryotic STING proteins bridge the gap between metazoans and bacteria.** (A) Graphical depiction of common domain architectures of STING proteins. (B) Maximum likelihood unrooted phylogenetic tree of STING domains from Metazoa and bacteria, which are separated by 1 long branch. Black dot (•) indicates proteins that have been previously experimentally characterized. Bacterial sequences are in gray and animal sequences are in red. (C) Maximum likelihood unrooted phylogenetic tree of hits from iterative HMM searches for diverse eukaryotic STING domains. The STING domains from blSTINGs from diverse eukaryotes break up the long branch between bacterial and animal STINGs. Structures of the indicated STING proteins are shown above, with those predicted by AlphaFold indicated by an asterisk. Homologs with X-ray crystal structures are from [3,87]. Two domain architectures exist in bacteria and eukaryotes (STING linked to a TIR domain and STING linked to a transmembrane domain), each of which have evolved convergently through domain shuffling. Ultrafast bootstraps determined by IQtree shown at key nodes. Eukaryotic sequences are colored according to the eukaryotic group as in Fig

1B. See S5 Fig for full STING phylogenetic tree. Underlying Newick files are included in Supporting information (S3 File and S4 File). AlphaFold predicted structures are also included in the Supporting information (S6 File, S7 File, and S8 File). blSTING, bacteria-like STING; HMM, hidden Markov model; STING, Stimulator of Interferon Genes.

that the metazoan sequences were separated from the bacterial sequences by one very long branch, along which many changes had occurred (Fig 3B).

To improve the phylogeny through the inclusion of a greater diversity of eukaryotic STING sequences, we began by carefully identifying the region of STING that was homologous between bacterial and animal STINGs, as we expected this region to be best conserved across diverse eukaryotes. Although Pfam domain PF15009 (TMEM173) is commonly used to define animal STING domains, this HMM includes a portion of STING's transmembrane domain which is not shared by bacterial STINGs. Therefore, we compared the crystal structures of HsSTING (6NT5), *Flavobacteriaceae* sp. STING (6WT4), and *Crassostrea gigas* STING (6WT7) to define a core "STING" domain. We used the region corresponding to residues 145–353 of 6NT5 as an initial HMM seed alignment of 15 STING sequences from PF1500915 ("Reviewed" sequences on InterPro). Our searches yielded 146 eukaryotic sequences from 64 species, which included STING homologs from 34 metazoans (S31 File and Fig 1). Using maximum likelihood phylogenetic reconstruction on the STING domain alone, we identified STING-like sequences from 26 diverse microeukaryotes whose STING domains clustered in between bacterial and metazoan sequences, breaking up the long branch. We name these sequences the bacteria-like STINGs (blSTINGs) because they were the only eukaryotic group of STINGs with a bacteria-like Prok_STING domain (PF20300) and because of the short branch length (0.86 versus 1.8) separating them from bacterial STINGs on the tree (Fig 3C). While a previous study reported STING domains in 2 eukaryotic species (1 in Stramenopiles and 1 in Haptista) [19], we were able to expand this set to additional species and also recover blSTINGs from Amoebozoa, Rhizaria, and choanoflagellates. This diversity allowed us to place the sequences on the tree with high confidence (bootstrap value >70), recovering a substantially different tree than previous work [19]. As for CD-NTases, the tree topology we recovered was robust across multiple different alignment and phylogenetic tree construction algorithms (S3A Fig).

Given the similarities between the STING domains of the blSTINGs and bacterial STINGs, we next asked whether the domain architectures of these proteins were similar using Hmmscan and AlphaFold. The majority of the new eukaryotic blSTINGs were predicted to have 4 N-terminal alpha helices (Fig 3C and S5 File and S6 File), similar to human STING. While bacterial TM-STINGs had superficially similar N-terminal transmembrane domains, these proteins were predicted to have only 2 alpha helices and in all phylogenetic trees the STING domains from bacterial TM-STINGs were more similar to other bacterial STINGs than to eukaryotic homologs (S3A Fig). These results suggest that eukaryotes and bacteria independently converged on a common TM-STING domain architecture through domain shuffling.

Interestingly, a similar pattern of convergent domain shuffling appears to have occurred a second time with the TIR-STING proteins. Some eukaryotes, such as the oyster *C. gigas*, have a TIR-STING fusion protein [3,51,59]. The STING domain of these TIR-STINGs clustered closely to other metazoan STINGs, suggesting an animal origin (Fig 3B). We also investigated the possibility that *C. gigas* acquired the TIR-domain of its TIR-STING protein via HGT from bacteria; however, this analysis also suggested an animal origin for the TIR domain (S7 Fig), as the *C. gigas* TIR domain clustered with other metazoan TIR domains such as *Homo sapiens* TICAM1 and 2 (ultrafast bootstrap value of 75). Eukaryotic TIR-STINGs are also rare, further

supporting the hypothesis that this protein resulted from recent convergence, where animals independently fused STING and TIR domains to make a protein resembling bacterial TIR-ST-INGs, consistent with previous reports [19]. Overall, the phylogenetic tree we constructed (Fig 3C) suggests that there is domain-level homology between bacterial and eukaryotic STINGs, but due to sparseness and lack of a suitable outgroup, this tree does not definitively explain the eukaryotic origin of the STING domain. However, the data does clearly support a model in which convergent domain shuffling in eukaryotes and bacteria generated similar TM-STING and TIR-STING proteins independently. Interestingly, the non-metazoan, blSTINGs (Fig 3C) that are found in the Stramenopiles, Haptista, Rhizaria, Choanoflagellates, and Amoebozoa have a TM-STING domain architecture similar to animal STINGs but a STING domain more similar to bacterial STINGs.

## Viperin is an ancient and widespread immune family

Viperins are innate immune proteins that restrict the replication of a diverse array of viruses by conversion of nucleotides into 3′-deoxy-3′,4′didehdro- (ddh) nucleotides [4,15–17]. Incorporation of these ddh nucleotides into a nascent RNA molecule leads to chain termination, blocking RNA synthesis and inhibiting viral replication [15,18]. While metazoan viperin specifically catalyzes CTP to ddhCTP [15], homologs from archaea and bacteria can generate ddhCTP, ddhGTP, and ddhUTP [4,60]. Previous structural and phylogenetic analysis showed that eukaryotic viperins are highly conserved at both the sequence and structural level and that, phylogenetically, animal and fungal viperins form a distinct monophyletic clade compared to bacterial viperins [4,16,60].

As viperin proteins consist of a single Radical SAM protein domain, we iteratively searched EukProt beginning with domain PF04055 (Radical_SAM). The 194 viperin-like proteins we recovered came from 158 species spanning the full range of eukaryotic diversity, including organisms from all of the major eukaryotic supergroups, as well as some orphan taxa whose taxonomy remains open to debate (Fig 1, Ancyromonadida, Hemimastigophora, Malawimonadida). When we constructed phylogenetic trees from these sequences, we found that the large majority of the eukaryotic viperins cluster together in a single, monophyletic clade, separate from bacterial or archaeal viperins (Fig 4). Within the eukaryotic viperin clade, sequences from more closely related eukaryotes often clustered together (Fig 4, colored blocks), as would be expected if viperins were present and vertically inherited within eukaryotes for an extended period of time. The vast species diversity and tree topology both strongly support the inference that viperins are a truly ancient immune module and have been present within the eukaryotic lineage likely dating back to the LECA.

In addition to this deep eukaryotic ancestry, we also uncovered 2 examples of bacteria–eukaryote HGT that have occurred much more recently, both in Chloroplastida, a group within Archaeplastida. The first of these consists of a small clade of Archaeplastida (Clade A) consisting of marine algae such as *Chloroclados australicus* and *Nemeris dumetosa*. These algal viperins cluster closely with the marine cyanobacteria *Anabaena cylindrica* and *Plankthriodies* (Figs 4 and S6). The second clade (Clade B) includes 4 other Archaeplastida green algal species, mostly *Chlamydomonas* spp. In some of our trees, the Clade B viperins branched near to eukaryotic sequences from other eukaryotic supergroups; however, the placement of the neighboring eukaryotic sequences varied depending on the algorithms we used; only the Archaeplastida placement was consistent (Figs 4 and S3A and S6). Taken together, we conclude that viperins represent a class of ancient immune proteins that have likely been present in eukaryotes since the LECA. Yet, we also find ongoing evolutionary innovation in viperins via HGT, both among eukaryotes and between eukaryotes and bacteria.

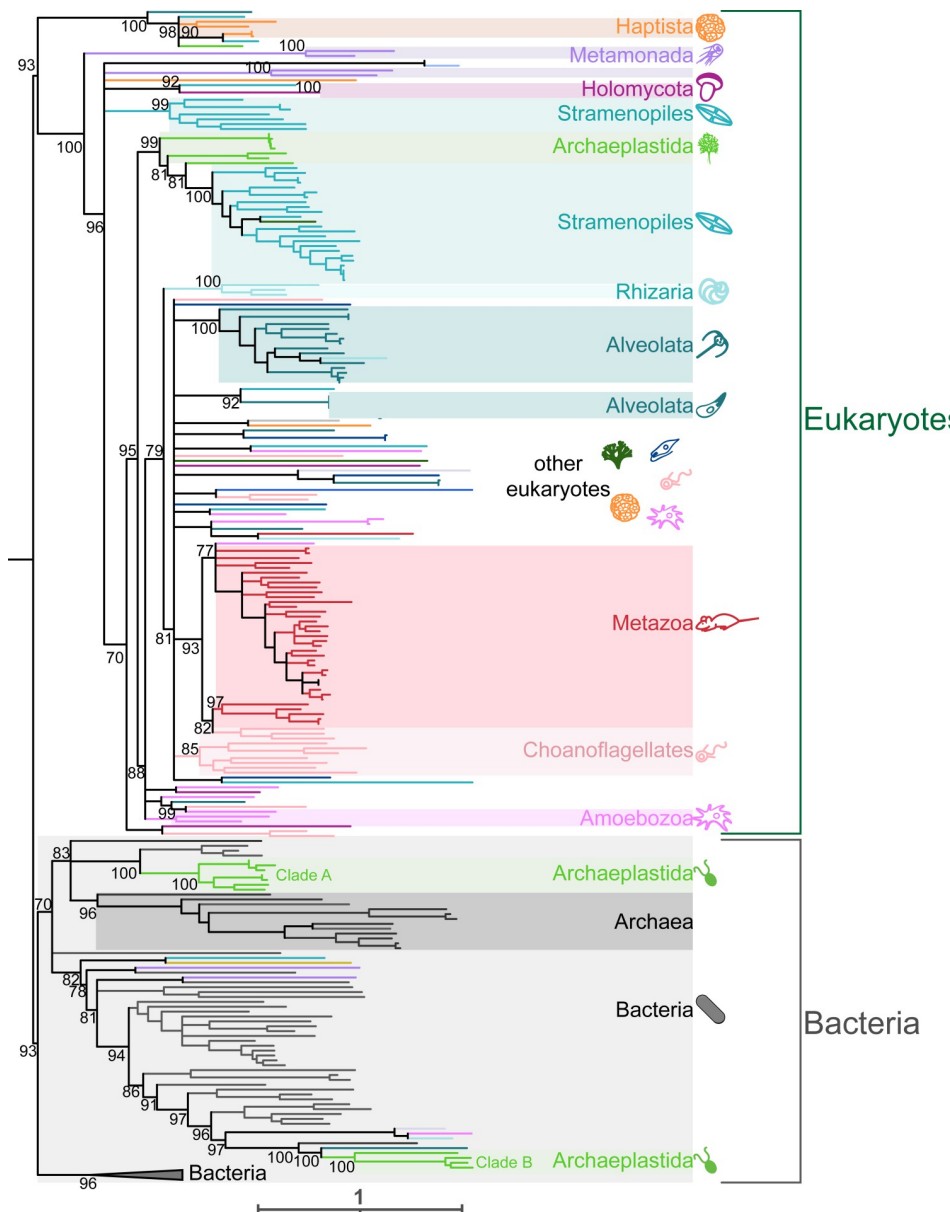

**Fig 4. Viperin is a deeply conserved innate immune module.** Maximum likelihood phylogenetic tree generated by IQtree of viperins from eukaryotes, bacteria, and archaea. All major eukaryotic supergroups have at least 2 species that encode a viperin homolog (colored supergroups). Eukaryotic sequences are colored according to eukaryotic group as in Fig 1B. Bacterial viperin sequences shown in gray and archaeal sequences in dark gray. There are 2 clades of Chloroplastida (a group within Archaeplastida) sequences that branch robustly (>80 ultrafast bootstrap value) within the bacteria clade. Ultrafast bootstraps determined by IQtree shown at key nodes. Tree is arbitrarily rooted between the major eukaryotic and bacterial clades. See S6 Fig for fully annotated viperin phylogenetic tree. Underlying Newick file is included in S8 File under Supporting information.

## Discussion

The recent discoveries that bacteria and mammals share mechanisms of innate immunity have been surprising, because they imply that there are similarities in immunity that span the Tree of Life. But how did these similarities come to exist? Here, we uncover several evolutionary trajectories that have led animals and bacteria to share homologous immune proteins

(summarized in Fig 5). We found that viperin dates back to at least the LECA and likely further. This finding has been recently confirmed through 2 studies that extend viperin history through Archaea [61,62]. We also uncovered examples of convergence, as in STING, where the shuffling of ancient domains has led animals and bacteria to independently arrive at similar protein architectures. Finally, we found evidence of multiple examples of bacteria–eukaryote HGTs that have given rise to immune protein families. An essential part of our ability to make these discoveries was the analysis of data from nearly 1,000 diverse eukaryotic taxa. These organisms allowed us to distinguish between proteins found across eukaryotes versus animal-specific innovations, to document both recent and ancient HGT events from bacteria that gave rise to eukaryotic immune protein families (Figs 2 and 4), and to identify STING proteins with eukaryotic domain architectures but more bacteria-like domains (blSTINGs, Fig 3). Because these diverged eukaryotic STINGs were found in organisms where we typically did not find any CD-NTase proteins, we hypothesize that blSTINGs may detect and respond to exogenous cyclic nucleotides, such as those generated by pathogens. In contrast to the STINGs, the eukaryotic CD-NTases had substantially different evolutionary histories, with multiple major CD-NTase superfamilies each emerging from within larger bacterial clades. While these analyses cannot definitively determine the directionality of the transfer, we favor the most parsimonious explanation that these components came into the eukaryotic lineage from bacterial origins.

While not as prevalent as in bacteria, HGT in eukaryotes represents a significant force in evolution, especially for unicellular species [63–66]. In this study, our criteria for "calling" HGT events was relatively strict, meaning that our estimate of HGT events is almost certainly an underestimate. Importantly, this pattern suggests that the bacterial pan-genome has been a rich reservoir that eukaryotes have repeatedly sampled to acquire novel innate immune components. Some of these HGT events have given rise to new eukaryotic superfamilies (e.g., eSMODS) that have never been characterized and could represent novel types of eukaryotic immune proteins. We speculate that the eSMODS superfamily CD-NTases and the blSTINGs may function more similarly to their bacterial homologs, potentially producing and responding to a variety of cyclic di- or tri-nucleotides [11]. Similarly, bacterial viperins have been shown to generate ddhCTP, ddhGTP, and ddhUTP, whereas animal viperins only make ddhCTP [4,15,60]. Thus, the 2 algal viperin clades arising from HGT may have expanded functional capabilities as well. A caveat of this work is that such strictly bioinformatic investigations are insufficient to reveal protein biochemical functions nor can they determine whether diverse homologs have been co-opted for non-immune functions. We therefore urge future, functional studies to focus on these proteins to resolve the questions of (1) whether/how blSTINGs operate in the absence of CD-NTases; (2) whether/how the functions of algal viperins and eSMODS changed following their acquisition from bacteria; and (3) whether the homologs truly function in immune defense.

In addition to these instances of gene gain, eukaryotic gene repertoires have been dramatically shaped by losses. Even for viperins, which likely date back to the eukaryotic last common ancestor, these proteins were sparsely distributed across eukaryotes and were absent from the majority of species we surveyed. While some of this finding may be due to technical limitations, such as dataset incompleteness or inability of the HMMs to recover distant homologs, we believe this explanation is insufficient to fully explain the sparseness, as many plant, fungal, and amoebozoan species are represented by well-assembled genomes where these proteins are certifiably absent (S2 Fig). Instead, we propose that the sparse distribution likely arises from ongoing and repeated gene loss, as has been previously documented for other gene families across the eukaryotic Tree of Life [28,30–32].

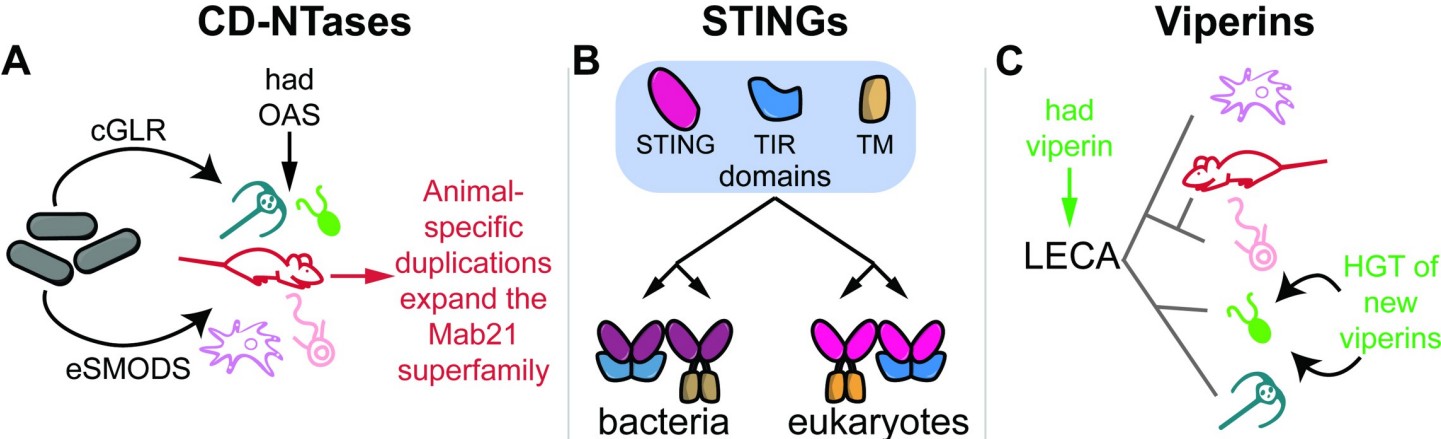

**Fig 5. Proposed model of evolutionary history of CD-NTases, STING, and viperin.** Summary of the proposed evolutionary history of each innate immune gene family. (A) We define 2 distinct superfamilies of CD-NTases that likely arose from bacteria–eukaryote HGT: eSMODS and cGLRs. Within the cGLR superfamily (which contains cGAS), a number of animal-specific duplications gave rise to numerous paralogs. The OAS superfamily of CD-NTases are abundant across diverse eukaryotic taxa and were likely present in the LECA. (B) Drawing on a shared ancient repertoire of protein domains that includes STING, TIR, and transmembrane (TM) domains, bacteria and eukaryotes have convergently evolved similar STING proteins through domain shuffling. (C) Viperins are widespread across the eukaryotic tree and likely were present in the LECA. In addition, 2 sets of recent HGT events from bacteria have equipped algal species with new viperins. CD-NTase, cGAS-DncV-like nucleotidyltransferase; HGT, horizontal gene transfer; LECA, last eukaryotic common ancestor; STING, Stimulator of Interferon Genes.

Overall, our results yield a highly dynamic picture of immune protein evolution across eukaryotes, wherein multiple mechanisms of gene gain are offset by ongoing losses. Interestingly, this pattern mirrors the sparse distributions of many of these immune homologs across bacteria [67–69], as antiphage proteins tend to be rapidly gained and lost from genomic defense islands [70,71]. It will be interesting to see if some eukaryotes evolve their immune genes in similarly dynamic islands, particularly in unicellular eukaryotes that undergo more frequent HGT [72].

We expect that our examination of CD-NTases, STING, and viperin represents just the tip of the iceberg when it comes to the evolution of eukaryotic innate immunity. New links between bacterial and animal immunity continue to be discovered and other immune families and domains such as Argonaute, Gasdermins, NACHT domains, CARD domains, TIR domains, and SamHD1 have been shown to have bacterial roots [2,6,7,9,10]. To date, the majority of studies have focused on proteins specifically shared between metazoans and bacteria. We speculate that there are probably many other immune components shared between bacteria and eukaryotes outside of animals. Further studies of immune defenses in microeukaryotes are likely to uncover new mechanisms of cellular defense and to better illustrate the origins and evolution of eukaryotic innate immunity.

## Materials and methods

### Iterative HMM search

The goal of this work was to search the breadth of EukProt v3 for immune proteins from the CD-NTase, STING, and viperin families that span the gap between metazoan and bacterial immunity. Our overall strategy was to first search with eukaryotes alone (starting from mainly Metazoa). Then, we added in bacterial sequences and searched with a mixed bacterial-eukaryotic HMM search until we either found no new hits, or until we began getting hits from an outgroup gene family. As outgroup sequences, we used Poly(A) RNA polymerase (PAP) sequences for the CD-NTases and molybdenum cofactor biosynthetic enzyme (MoaA) for

viperin. We did not have a suitable outgroup for STING domains nor did any diverged outgroups come up in our searches. In parallel to our searches spanning bacterial and eukaryotic protein diversity, we also performed bacteria-only and eukaryote-only searches to ensure that we found as many homologs as possible (schematized in Fig 1A and further in S1A Fig).

**Phase 1: Eukaryotic searches.** To begin, HMM profiles from Pfam (for CD-NTases and viperin) or an HMM profile generated from a multiple sequence alignment (for STING) were used to search EukProt V3 [25] for diverse eukaryotic sequences. For CD-NTases and viperin, HMM profiles of Pfams PF03281 and PF04055 were used respectively.

For STING, where the Pfam profile includes regions of the protein outside of the STING domain, we generated a new HMM for the initial search. First, we aligned crystal structures of HsSTING (6NT5), *Flavobacteriaceae* sp. STING (6WT4), and *Crassostrea gigas* STING (6WT7) with the RCSB PDB "Pairwise Structure Alignment" tool with a jFATCAT (rigid) option [73,74]. We defined a core "STING" domain, as the ungapped region of 6NT5 that aligned with 6WT7 and 6WT4 (residues G152-V329 of 6NT5). Then, we aligned 15 eukaryotic sequences from PF15009 (all 15 of the "Reviewed" sequences on InterPro) with MAFFT (v7.4.71) [75] with default parameters and manually trimmed the sequences down to the boundaries defined by our structural alignment (residues 145–353 of 6NT5). We then trimmed the alignment with TrimAI (v1.2) [76] with options -gt 0.2. The trimmed MSA was then used to generate an HMM profile with hmmbuild from the hmmer (v3.2.1) package (hmmer.org) using default settings. We ran these HMM searches of EukProtV3 with the script "wrap_hmmscan.pl", which searches each individual species file in EukProt and combines the results. This code, by Dan Richter, is available at https://github.com/MBL-Physiology-Bioinformatics/2021-Bioinformatics-Tutorial-Materials/tree/master/phylogenetics.

HMM profiles were used to search EukProt via hmmsearch (also from hmmer v3.2.1) with a statistical cutoff value of 1e-3 and -hit parameter set to 10 (i.e., the contribution of a single species to the output list is capped at 10 sequences). It was necessary to cap the output list, as EukProt v3 includes de novo transcriptome assemblies with multiple splice isoforms of the same gene and we wanted to limit the overall influence a single species had on the overall tree. We never reached the 10 sequence cap for any search for STING or viperin homologs; only for the CD-NTases within Metazoa did this search cap limit hits. The resulting sequences from this search were then aligned with hmmalign (included within hmmer) using settings "—outformat afa—trim [Protein.hmm]", where [Protein.hmm] is the HMM profile file that was used to do the previous search. This HMM alignment was then used to generate a new HMM profile with hmmbuild. This profile was used to search EukProt v3 again and the process was repeated until no new sequences were found or until sequences from other gene families were found, which was between 3 and 4 eukaryotic searches for all 3 protein families.

**Phase 2: Combining eukaryotic and bacterial sequences into an HMM.** After the eukaryotic searches reached saturation (i.e., no additional eukaryotic sequences were recovered after additional searches), bacterial sequences were acquired from previous literature (viperins from [4], CD-NTases from [11], and STINGs from [3,8,21]). To ensure the combined HMM did not have an overrepresentation of either bacterial or eukaryotic sequences, we downsampled the bacterial sequences and eukaryotic sequences to obtain 50 phylogenetically diverse sequences of each, and then combined the 2 downsampled lists. To do this, eukaryotic and bacterial sequences were each separately aligned with MAFFT (default parameters), phylogenetic trees were built with FastTree (v2.1.10) [77], and the Phylogenetic Diversity Analyzer (pda/1.0.3) [78] software with options -k 50 or -k 500 with otherwise default parameters was run the FastTree files to downsample the sequences while maximizing remaining sequence diversity.

The combined bacterial-eukaryotic sequence list was then aligned with hmmalign (with settings "—outformat afa—trim [Protein.hmm]") and used to construct a new HMM profile with hmmbuild (default parameters). This HMM profile was used to search EukProt v3 with settings -evalue 1e-3, and -hit 10. The eukaryotic hits from this search were then aligned with MAFFT (default parameters), and a tree was constructed with FastTree (default parameters). From this tree, the sequences were then downsampled with PDA (-k 50) and once again combined with the bacterial list, aligned, used to generate a new HMM, and a new search. This process was iterated 3 to 5 times until saturation or until the resulting sequence hits included other gene families that branched outside of the sequence diversity defined by the metazoan and bacterial homologs. See S26, S27 and S28 Files under Supporting information of the final HMMs from the CD-NTases, STING, and viperin, respectively.

**Phase 3: Searching with a bacteria-only sequences or existing HMM profiles.** To search EukProt v3 with a bacteria-only HMM for each protein family, we aligned the full set of published bacterial sequences with MAFFT (default parameters), trimmed with TrimAI (-gt 0.2), and hmmbuild (default parameters) was used to generate an HMM profile that was used to search EukProt v3. As a point of comparison, we also searched the database with only the starting, previously constructed Pfam HMMs for CD-NTases (PF03281), STING (PF15009), and viperin (PF04055).

**Phase 4: Combining all hits into a single list and scanning for domains.** Sequences from all iterative searches were combined to generate a total hits FASTA file for STING, CD-NTase, and viperin. First, duplicate sequences were removed, then the fasta files were scanned using hmmscan (also from hmmer v3.2.1) with settings "—domtblout—domE 1e-3" against the Pfam database (Pfam-A.hmm) and all predicted domains with an E-value <1e-3 were considered. Next, we generated phylogenetic trees (first by aligning with MAFFT (default parameters) and then building a tree with FastTree) and used these trees along with the hmmscan domains to determine in-group and out-group sequences. Out-group sequences were manually removed from the fasta file. We determined outgroup sequences by these criteria: (1) if the sequence clustered outside of known outgroup sequences (e.g., PAP sequences for the CD-NTases and molybdenum cofactor biosynthetic enzyme (MoaA) for viperin); or (2) if sequence did not have at least one of the relevant domains (Mab21/OAS1-C/SMODS for CD-NTases, TMEM173/Prok_STING for STING, and Radical_SAM for viperin). These 3 FASTA files were used for the final alignments and phylogenetic trees. To identify protein domains in each sequence, the FASTA files were scanned using hmmscan (also from hmmer v3.2.1) against the Pfam database (Pfam-A.hmm) and all predicted domains with an E-value <1e-3 were considered. See S25 File for the hmmscan results of all included homologs.

## Final alignment and tree building

To generate final phylogenetic trees, all eukaryotic search hits and bacterial sequences were aligned using MAFFT (default parameters). We downsampled the CD-NTase bacterial sequences from approximately 6,000 down to 500 using PDA software (options -k 500) on a FastTree (default settings) tree built upon an MAFFT (default parameters) tree, to facilitate more manageable computation times on alignments and tree construction. For the STING and viperin trees, we included all bacterial sequences. These initial alignments were first trimmed manually in Geneious (v2023.1.2) to remove unaligned N- and C-terminal regions, and then realigned with MAFFT (default parameters) or MUSCLE (v5.1) [79] and trimmed with TrimAI (v1.2) [76]. MUSCLE was used with the "-super5" option and otherwise default parameters. MUSCLE was deployed in parallel with MAFFT to generate these final alignments to ensure that the final tree topology would be as robust as possible. MUSCLE is a slightly

more accurate but more computationally intensive alignment software [79]. The length of these final alignments were 232, 175, and 346 amino acids long for CD-NTase, STING, and viperin domains, respectively. These alignments represent ≥75% of the length of alignment their respective PFAM domain (PF3281 (Mab-21 protein nucleotidyltransferase domain) for CD-NTases, PF20300 (Prokaryotic STING domain) for STING, and PF404055 (Radical SAM family) for viperin). These alignments were used to generate phylogenetic trees using 3 tree inference softwares: FastTree (v2.1.10) [77], IQtree (2.2.2.7) [80], and RaxML-ng (v0.9.0) [81]. FastTree was utilized with default settings. IQtree was used to determine the appropriate evolutionary model and was run with 1,000 ultrafast bootstraps (IQtree settings: -s, -bb 1000, -m TEST, -nt AUTO). RaxML-ng trees were produced with 100 bootstraps using the molecular model specified from the IQtree analysis (Raxml-ng settings:—all,—model [specified by IQtree],—tree pars{10}—bs-trees 100). Phylogenetic trees were visualized with iTOL [82]. Weighted Robinson–Foulds distances for S3B Fig were calculated with Visual TreeCmp (settings: -RFWeighted -Prune trees -include summary -zero weights allowed) [83].

### TIR domain alignment and tree

We used hmmscan to identify the coordinates of TIR domains in a list of 203 TIR domain containing-sequences from InterPro (all 203 proteins from curated "Reviewed" selection of IPR000157 (Toll/interleukin-1 receptor homology (TIR) domain as of 2023-04-04)) and 104 bacterial TIR-STING proteins (the same TIR-STING proteins used in Fig 3) [3]. Next, we trimmed the sequences down to the hmmscan identified TIR coordinates and aligned the TIR domains with MUSCLE (-super5). We trimmed the alignments with TrimAL and built a phylogenetic tree with IQtree (-s, -bb 1000, -m TEST, -nt AUTO).

### Venn diagrams

Venn diagrams were generated via DeepVenn [84] using presence/absence information for cGLR, OAS, and STING from each eukaryotic species that encoded at least one of these proteins.

### Protein structure modeling

To model 3D protein structures for STING homologs without a published crystal structure, we ran AlphaFold (v2.1.1) [85,86]. We generated 5 ranked models for STINGs from *Flavobacteriaceae* (IMG ID: 2624319773), *Nitzschia* sp. (EukProt ID: P007051), and *Caveostelium apophsatum* (EukProt ID: P019191). Fig 2C shows highest ranked models only. These highest ranked models are provided as S5, S6 and S7 Files under Supporting information for Nitzschia, Caveostelium, and Flavobacteriaceae, respectively.

### Supporting information

**S1 Fig. Collectors curves and full search strategy.** (A) Detailed schematic outlining the iterative HMM search strategy. Blue boxes and blue shaded region show eukaryotic-only searches to create pan-eukaryotic HMMs and yellow indicates eukaryotic-bacterial searches to create universal HMMs. For the combined bacterial/eukaryotic searches (yellow box), bacterial and eukaryotic sequences were each downsampled to 50 sequences (phylogenetic tree downsampled via PDA) to maintain equal contributions from bacteria and eukaryotic sequences. Separately, bacterial sequences were aligned and used to make an HMM which was used to search EukProt as a "bacteria only search" and for STING we searched with PF15009 for a comparable Eukaryotic PFAM search (not shown in flowchart). We did this extra search for

STING as PF15009 contains part of the eukaryotic STING transmembrane domain and so our first search with STING was with a STING-domain-only HMM (see Materials and methods). Pink (MUSCLE) and orange (MAFFT) boxes show the final alignments and phylogenetic trees that were constructed. (B) STING, CD-NTase, and viperin collectors curves showing the number of cumulative protein sequences that were found after each iterative search. Results from eukaryotic searches are shown in blue and the combined searches in yellow. Solid black line indicates the number of hits from the starting Pfam HMM alone and the dotted gray line shows the number of hits from a bacteria-only HMM. Note that some searches yielded hits that were members of more distant protein families, which were later removed from the analysis and are not counted here. Individual data are available in S1 File.
(TIF)

**S2 Fig. Data quality of EukProt species by data type.** Species trees representing organisms included in EukProt v3 as genomes (A) or transcriptomes (B). Supergroups are color coded as in Fig 1B. Colored bars mark each eukaryotic species in which the HMM search found a homolog sequence of STING, CD-NTase, or Viperin. Black bar chart shows BUSCO completeness score for each genome/transcriptome, with higher bars indicating higher data set completeness. BUSCO scores can also be viewed on EukProt v3 (https://evocellbio.com/SAGdb/images/EukProtv3.busco.output.txt). Individual data are included in S1 File.
(TIF)

**S3 Fig. Phylogenetic trees from different alignments and tree building methods show robust topologies.** (A) Unrooted maximum likelihood phylogenetic trees generated from 2 separate alignments (MUSCLE and MAFFT) and with 2 different tree inference programs (IQtree and RaxML-ng). Scale bar of 1 shown beneath each tree represents the number of amino acid substitutions per position in the underlying alignment. Colored branches show eukaryotic sequences with the same color scheme as Fig 1B, while gray lines are bacterial sequences. For the majority of relationships discussed here, we recovered the same tree topology at key nodes regardless of alignment or tree reconstruction algorithm used. (B) The weighted Robinson–Foulds distances all pairwise comparisons between the 4 tree types (MAFFT/MUSCLE alignment built with IQTREE/RAXML-ng). Although the distances were higher for the CD-NTase tree (as expected for this highly diverse gene family), all of the key nodes defining the cGLR, OAS, and eSMODS superfamilies, as well as their nearest bacterial relatives, were well supported (>70 ultrafast bootstrap value). Underlying alignment and Newick files are included (Alignments: S9, S10, S11, S12, S13, S14 Files. Newick files: S2, S4, S8, S15, S16, S17, S18, S19, S20, S21, S22, S23 Files) under Supporting information. All pairwise comparisons for weighted Robinson–Foulds distance calculations are included in S1 File.
(TIF)

**S4 Fig. CD-NTase phylogenetic tree.** Maximum likelihood phylogenetic tree generated by IQtree of hits from iterative HMM searches for diverse eukaryotic CD-NTases. Tree is arbitrarily rooted between bacterial CD-NTase clades. Scale bar represents the number of amino acid substitutions per position in the underlying MUSCLE alignment. Eukaryotic sequences are color coded as in Fig 1B. Ultrafast bootstrap values calculated by IQtree at all nodes with support >70 are shown. Branches with support values <70 were collapsed to polytomies. Underlying Newick file is included in S2 File under Supporting information.
(TIF)

**S5 Fig. STING phylogenetic tree.** Maximum likelihood phylogenetic tree of hits from iterative HMM searches for diverse eukaryotic STING domains. Tree is arbitrarily rooted on a branch separating the bacterial sequences from eukaryotes. Scale bar represents the number of amino

acid substitutions per position in the underlying MUSCLE alignment. Eukaryotic sequences are color coded as in Fig 1B. Ultrafast bootstrap values calculated by IQtree at all nodes with support >70 are shown. Branches with support values <70 were collapsed to polytomies. Underlying Newick file is included in S4 File under Supporting information.
(TIF)

**S6 Fig. Viperin phylogenetic tree.** Maximum likelihood phylogenetic tree generated by IQtree of hits from iterative HMM searches for diverse eukaryotic viperins. Tree is arbitrarily rooted on a branch separating the bacterial sequences from eukaryotes. Scale bar represents the number of amino acid substitutions per position in the underlying MUSCLE alignment. Eukaryotic sequences are color coded as in Fig 1B. Ultrafast bootstrap values calculated by IQtree at all nodes with support >70 are shown. Branches with support values <70 were collapsed to polytomies. Underlying Newick file is included in S8 File under Supporting information.
(TIF)

**S7 Fig. TIR domain of *Crassostrea gigas'* TIR-STING is closely related to metazoan TIR domains.** Unrooted maximum likelihood tree of diverse TIR domains. Scale bars on the phylogenetic tree represent the number of amino acid substitutions per position in the underlying MUSCLE alignment. Eukaryotic sequences are color coded as in Fig 1B. Ultrafast bootstrap values calculated by IQtree at key nodes are shown. Underlying Newick file is included in S24 File under Supporting information.
(TIF)

**S1 File A. xlsx file with 3 tabs: Catalogs, Collectors Curves, Venn Diagram, and Robinson–Foulds.** The Catalogs tab has the EukProt Species IDs and whether a homolog was found (1 = found homolog, 0 = did not find homolog), for each protein family. This tab makes up the raw data from which Figs 1B and S2 were generated. The Collectors Curves tab has the raw data used to make the graphs for S1B Fig. The number of search hits for each specified search is enumerated for each protein family. Searches that were not carried out are blank. The Venn Diagram tab has the EukProt Species ID against the presence/absence of a given homolog in Metazoa and non-metazoans (1 = found homolog, 0 = did not find homolog). The Robinson–Foulds tab has the raw data for each pairwise comparison between the various phylogenetic trees.
(XLSX)

**S2 File. Newick file of maximum likelihood phylogenetic tree of CD-NTases generated from a MUSCLE (v5.1) (S10 File) alignment with IQtree (2.2.2.7).** Newick file is used in Figs 2, S3A and S4. Node support values calculated from ultrafast bootstraps.
(TREE)

**S3 File. Newick file of maximum likelihood phylogenetic tree of animal and bacterial STING domains generated from a MUSCLE (v5.1) alignment with IQtree (2.2.2.7).** Newick file is used in Fig 3B.
(TREE)

**S4 File. Newick file of maximum likelihood phylogenetic tree of STING domains generated from a MUSCLE (v5.1) (S12 File) alignment with IQtree (2.2.2.7).** Newick file is used in Figs 3C, S3A and S5. Node support values calculated from ultrafast bootstraps.
(TREE)

**S5 File. A protein structure predicted by AlphaFold of EP00394_Nitzschia_sp_-Nitz4_P007501.** This .pdb structure was predicted as a dimer and is used in Fig 3C.
(PDB)

**S6 File. A protein structure predicted by AlphaFold of EP01114_Cavostelium_apophysatum_P019191.** This .pdb structure was predicted as a dimer and is used in Fig 3C.
(PDB)

**S7 File. A protein structure predicted by AlphaFold of Flavobacteriaceae STING (IMG ID: 2624319773).** This .pdb structure was predicted as a dimer and is used in Fig 3C.
(PDB)

**S8 File. Newick file of maximum likelihood phylogenetic tree of viperins generated from a MUSCLE (v5.1) (S14 File) alignment with IQtree (2.2.2.7).** Newick file is used in Figs 4, S3 and S6. Node support values calculated from ultrafast bootstraps.
(TREE)

**S9 File. FASTA file of an MAFFT (v7.4.71) alignment for CD-NTases.** This MAFFT alignment was used to construct phylogenetic trees for S3 Fig.
(FASTA)

**S10 File. FASTA file of a MUSCLE (v5.1) alignment for CD-NTases.** This MUSCLE alignment was used to construct phylogenetic trees for Figs 2, S3 and S4.
(FASTA)

**S11 File. FASTA file of an MAFFT (v7.4.71) alignment for STING.** This MAFFT alignment was used to construct phylogenetic trees for S3 Fig.
(FASTA)

**S12 File. FASTA file of a MUSCLE (v5.1) alignment for STING.** This MUSCLE alignment was used to construct phylogenetic trees for Figs 3, S3 and S5.
(FASTA)

**S13 File. FASTA file of an MAFFT (v7.4.71) alignment for viperin.** This MAFFT alignment was used to construct phylogenetic trees for S3 Fig.
(FASTA)

**S14 File. FASTA file of a MUSCLE (v5.1) alignment for viperin.** This MUSCLE alignment was used to construct phylogenetic trees for Figs 4, S3 and S6.
(FASTA)

**S15 File. Newick file of maximum likelihood phylogenetic tree of CD-NTases generated from an MAFFT alignment (S9 File) with IQtree (2.2.2.7).** Newick file is used in S3 Fig. Node support values calculated from ultrafast bootstraps.
(TREE)

**S16 File. Newick file of maximum likelihood phylogenetic tree of CD-NTases generated from an MAFFT alignment (S9 File) with RaxML-ng (v0.9.0).** Newick file is used in S3 Fig.
(TREE)

**S17 File. Newick file of maximum likelihood phylogenetic tree of CD-NTases generated from a MUSCLE alignment (S10 File) with RaxML-ng (v0.9.0).** Newick file is used in S3 Fig.
(TREE)

**S18 File. Newick file of maximum likelihood phylogenetic tree of STING domains generated from an MAFFT alignment (S11 File) with IQtree (2.2.2.7).** Newick file is used in S3

Fig. Node support values calculated from ultrafast bootstraps.
(TREE)

**S19 File. Newick file of maximum likelihood phylogenetic tree of STING domains generated from an MAFFT alignment (S11 File) with RaxML-ng (v0.9.0).** Newick file is used in S3 Fig. Node support values calculated from ultrafast bootstraps.
(TREE)

**S20 File. Newick file of maximum likelihood phylogenetic tree of STING domains generated from a MUSCLE alignment (S12 File) with RaxML-ng (v0.9.0).** Newick file is used in S3 Fig. Node support values calculated from ultrafast bootstraps.
(TREE)

**S21 File. Newick file of maximum likelihood phylogenetic tree of viperins generated from an MAFFT alignment (S13 File) with IQtree (2.2.2.7).** Newick file is used in S3 Fig. Node support values calculated from ultrafast bootstraps.
(TREE)

**S22 File. Newick file of maximum likelihood phylogenetic tree of viperins generated from an MAFFT alignment (S13 File) with RaxML-ng (v0.9.0).** Newick file is used in S3 Fig. Node support values calculated from ultrafast bootstraps.
(TREE)

**S23 File. Newick file of maximum likelihood phylogenetic tree of viperins generated from a MUSCLE alignment (S14 File) with RaxML-ng (v0.9.0).** Newick file is used in S3 Fig. Node support values calculated from ultrafast bootstraps.
(TREE)

**S24 File. Newick file of maximum likelihood phylogenetic tree of TIR domains generated from a MUSCLE alignment (S29 File) with IQtree (2.2.2.7).** Newick file is used in S7 Fig. Node support values calculated from ultrafast bootstraps.
(TREE)

**S25 File. A .xlsx file with Hmmscan data for each CD-NTase, STING, and viperin protein sequence found in Figs 2A, 3C and 4, respectively.** Each protein family is located on a different tab. Table headers include Query Name, Target Name, Target Length, E-Value, score, bias, Alignment Coordinate from:, Alignment Coordinate to:, and Description. These table headers are standard for Hmmscan and define how good of a match a domain in PFAM (a "Target") is to the protein in a list (a "Query").
(XLSX)

**S26 File. Final HMM file for the CD-NTases.**
(HMM)

**S27 File. Final HMM file for STING.**
(HMM)

**S28 File. Final HMM file for viperin.**
(HMM)

**S29 File. FASTA file of a MUSCLE (v5.1) alignment for TIR domains.** This MUSCLE alignment was used to construct the phylogenetic tree for S7 Fig.
(FASTA)

**S30 File. Fasta file with all CD-NTase amino acid sequences analyzed.** This list is composed of all full-length sequences (both bacterial and eukaryotic) that make up Fig 2A.
(FASTA)

**S31 File. Fasta file with all STING amino acid sequences analyzed.** This list is composed of all full-length sequences (both bacterial and eukaryotic) that make up Fig 3C.
(FASTA)

**S32 File. Fasta file with all viperin amino acid sequences analyzed.** This list is composed of all full-length sequences (both bacterial and eukaryotic) that make up Fig 4.
(FASTA)

## Acknowledgments

We thank Daniel Richter for his feedback, encouragement, and scientific guidance. Maureen Stolzer, Kevin Forsberg, Patrick Mitchell, and members of the Levin lab also provided helpful input on the project and manuscript, as did three reviewers. Thanks to Andrew VanDemark for helpful discussions about 3-D modeling and to Jacob Durrant for help running AlphaFold.

## Author Contributions

**Conceptualization:** Tera C. Levin.

**Data curation:** Edward M. Culbertson.

**Formal analysis:** Edward M. Culbertson.

**Funding acquisition:** Tera C. Levin.

**Investigation:** Edward M. Culbertson, Tera C. Levin.

**Methodology:** Edward M. Culbertson.

**Project administration:** Edward M. Culbertson, Tera C. Levin.

**Supervision:** Tera C. Levin.

**Validation:** Edward M. Culbertson, Tera C. Levin.

**Visualization:** Edward M. Culbertson, Tera C. Levin.

**Writing – original draft:** Edward M. Culbertson, Tera C. Levin.

**Writing – review & editing:** Edward M. Culbertson, Tera C. Levin.

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
