## [Editor Report · Decision Letter 0]

10 Nov 2023

Dear Dr. Levin, 

Thank you for submitting your manuscript entitled "Eukaryotic antiviral immune proteins arose via domain shuffling, horizontal transfer, and ancient inheritance" for consideration as a Research Article by PLOS Biology.

Your manuscript, reviewers' comments from Review Commons and your responses to reviewers have now been evaluated by the PLOS Biology editorial staff and I am writing to let you know that we would like to consider your manuscript further.

However, before we can consider your manuscript further, we need you to complete your submission by providing the metadata that is required for full assessment. To this end, please login to Editorial Manager where you will find the paper in the 'Submissions Needing Revisions' folder on your homepage. Please click 'Revise Submission' from the Action Links and complete all additional questions in the submission questionnaire.

Once your full submission is complete, your paper will undergo a series of checks. To provide the metadata for your submission, please Login to Editorial Manager (https://www.editorialmanager.com/pbiology) within two working days, i.e. by Nov 12 2023 11:59PM.

Kind regards,

Paula

---

Senior Editor

PLOS Biology

---

## [Editor Report · Decision Letter 1]

14 Nov 2023

Dear Dr. Levin,

Thank you for your patience while your manuscript "Eukaryotic antiviral immune proteins arose via domain shuffling, horizontal transfer, and ancient inheritance" along with the reviews from Review Commons was being assessed at PLOS Biology. It has now been evaluated by the PLOS Biology editors and an Academic Editor with relevant expertise, and by several independent reviewers.

Based on our Academic Editor's assessment of your revision, we are likely to accept this manuscript for publication, provided you satisfactorily address the following data and other policy-related requests.

1. DATA POLICY:

A) Supplementary files (e.g., excel). Please ensure that all data files are uploaded as 'Supporting Information' and are invariably referred to (in the manuscript, figure legends, and the Description field when uploading your files) using the following format verbatim: S1 Data, S2 Data, etc. Multiple panels of a single or even several figures can be included as multiple sheets in one excel file that is saved using exactly the following convention: S1_Data.xlsx (using an underscore).

B) Deposition in a publicly available repository. Please also provide the accession code or a reviewer link so that we may view your data before publication.

Regardless of the method selected, please ensure that you provide the individual numerical values that underlie the summary data displayed in the following figure panels as they are essential for readers to assess your analysis and to reproduce it: Figures 1B, 2ABCD, 3BC, 4, and Supplementary Figures SF1B, SF2AB, SF3A, SF4, SF5, SF6, SF7.

**Please also ensure that figure legends in your manuscript include information on where the underlying data can be found, and ensure your supplemental data file/s has a legend.**

2. CODE POLICY

Per journal policy, as the code that you have generated is important to support the conclusions of your manuscript, we require that you make it available without restrictions upon publication. Please ensure that the code is sufficiently well documented and reusable, and that your Data Statement in the Editorial Manager submission system accurately describes where your code can be found.

3. We think the title is misleading - as currently written, it implies that all eukaryotic antiviral immune proteins arose via shuffling, transfer, and ancient inheritence, but the paper documents this for a limited set of eukaryotic proteins. We suggest a more specific title: "Eukaryotic antiviral immune proteins of the CD-NTase, STING and viperin families arose from bacteria via domain shuffling, horizontal transfer and ancient inheritance".

We expect to receive your revised manuscript within two weeks.

*Published Peer Review History*

*Press*

Sincerely,

Paula

---

Senior Editor,

pjaureguionieva@plos.org,

PLOS Biology

---

## [Editor Report · Decision Letter 2]

20 Nov 2023

Dear Dr Levin,

Thank you for the submission of your revised Research Article "Eukaryotic CD-NTase, STING and viperin proteins evolved via domain shuffling, horizontal transfer and ancient inheritance from prokaryotes" for publication in PLOS Biology. On behalf of my colleagues and the Academic Editor, Michael Laub, I am pleased to say that we can in principle accept your manuscript for publication, provided you address any remaining formatting and reporting issues. These will be detailed in an email you should receive within 2-3 business days from our colleagues in the journal operations team; no action is required from you until then. Please note that we will not be able to formally accept your manuscript and schedule it for publication until you have completed any requested changes.

PRESS

Sincerely, 

Paula

---

Senior Editor

PLOS Biology
